# An Improved Method Based on Bluetooth Low-Energy Fingerprinting for the Implementation of PEPS System

**DOI:** 10.3390/s22249615

**Published:** 2022-12-08

**Authors:** Francesco Bonavolontà, Annalisa Liccardo, Rosario Schiano Lo Moriello, Enzo Caputo, Giorgio de Alteriis, Angelo Palladino, Giuseppe Vitolo

**Affiliations:** 1Dipartimento di Ingegneria Elettrica e delle Tecnologie dell’Informazione, Università degli Studi di Napoli Federico II, 80125 Naples, Italy; 2Dipartimento di Ingegneria Industriale, Università degli Studi di Napoli Federico II, 80125 Naples, Italy; 3Kineton, Via Emanuele Gianturco 23, 80142 Naples, Italy

**Keywords:** PEPS, BLE, Kalman Filter, k-Nearest Neighbor

## Abstract

In the automotive field, the introduction of keyless access systems is revolutionizing car entry techniques currently dominated by a physical key. In this context, this paper investigates the possible use of smartphones to create a PEPS (Passive Entry Passive Start) system using the BLE (Bluetooth Low-Energy) Fingerprinting technique that allows, along with a connection to a low-cost BLE micro-controllers network, determining the driver’s position, either inside or outside the vehicle. Several issues have been taken into account to assure the reliability of the proposal; in particular, (i) spatial orientation of each microcontroller-based BLE node which ensures the best performance at 180° and 90° referred to as the BLE scanner and the advertiser, respectively; (ii) data filtering techniques based on Kalman Filter; and (iii) definition of new network topology, resulting from the merger of two standard network topologies. Particular attention has been paid to the selection of the appropriate measurement method capable of assuring the most reliable positioning results by means of the adoption of only six embedded BLE devices. This way, the global accuracy of the system reaches 98.5%, while minimum and maximum accuracy values relative to the individual zones equal, respectively, to 97.3% and 99.4% have been observed, thus confirming the capability of the proposed method of recognizing whether the driver is inside or outside the vehicle.

## 1. Introduction

Location tracking is one of the most valuable tools and/or goals for both industrial and academic fields. The leakage of GPS (Global Positioning System) signal in indoor or harsh environments as well as its poor location performance has led to research into and development of a series of technologies capable of overcoming such limitations and conceiving of new applications. Among them, BLE is a medium-range wireless communication technology, proposed to satisfy the need to reduce possible energy consumption, costs, bandwidth and power dissipation. Thanks to the considered features, this technology is taking over in IoT (Internet of Things) applications; a typical case of interest concerns human-to-machine interaction.

In the literature, there are several works using the BLE protocol to implement the indoor localization method; for example, in [1], the authors use a Fingerprint-based indoor localization method to locate a target device whose location is unknown where the authors propose a solution based on Principle Component Analysis (PCA) or Autoenconder (AE) to extract several Fingerprint feature. In [2], the BLE transmitter’s position was estimated through a Deep Neural Network-based indoor localization by the signal received from several Anchor Points. To train and test the proposed Convolutional Neural Network (CNN) a large dataset was obtained by means of a simulated environment. In [3], the authors present a method for simultaneous indoor pedestrian localization and house mapping through combined data from an Inertial Measurement Unit with proximity and activity data from Bluetooth Low-Energy beacons placed indoors deployed in the indoor environment. In [4], a system that locates and guides a user inside a building in real-time is presented. This smartphone application captures the Bluetooth Low-Energy signal from the beacons and displays the location of the user’s smartphone on a map that is obtained from Building Information Modeling (BIM), guiding the user to the desired destination. In [5], BLE protocol is used to define the position of a user and multi-users in indoor environments; as the first scenario, the authors propose a localization server, client and user application based on a trilateration and Center of Gravity (COG) calculation; a second research element was an algorithm to improve the localization performance by means of a range-average algorithm exploiting the inertial sensors. Concerning the automotive field, human-to-machine interaction is important to define the driver’s position with respect to the vehicle through the definition of a PEPS system. Typically, PEPS systems use the localization algorithm called *RF ranging*. Based on the assumption that radio waves propagate according to the inverse-square law [6], the distance can be estimated as a function of the transmitted and received signal strength, provided that there are no other sources of error falsifying the result. When applying localization algorithms in indoor environments, the signal strength is extremely noisy due to reflection or absorption phenomena with the structure [7]. In [8], the advantages of the BLE adoption compared to Wi-Fi are highlighted in an indoor application where the relation between the RSSI measurements and distance represent a crucial aspect to evaluate the node position; in fact, the transmission and reception node distance affect the RSSI signal power. Various radio frequency-based positioning algorithms can be applied to reduce this problem, including:Time of Arrival (ToA): The distance between the emitter and the receiving node is calculated considering the time elapsed between the emission and reception of the signal. Knowing the speed of propagation of the signal in the medium and the time it takes to arrive from one point to the other (ToF or Time of Flight), it is possible to calculate the traveled distance.Time Difference of Arrival (TDoA): This is based on the same principles as ToA. Differently from ToA, TDoA exploits the difference in time of flight between the emission and reception of the signal to calculate the distance between the reference node, whose coordinates are already known.Angle of Arrival (AoA): This determines the position of a mobile device through the angle at which the signal reaches two sensors of known coordinates. Then, applying the triangulation technique, it is possible to obtain the coordinates of the transmitter, which will be located at the intersection of the two directions.

Another localization algorithm is based on RF Fingerprinting [9]; this technique allows the exploitation of pre-existing access points within an environment in order to return the position of a device.

In [10], the authors introduced the BLE to realize a PEPS system. In particular, their interest was to reduce BLE’s power dissipation and they did not focus on localization accuracy. In [11], the authors propose an adaptive surrounding BLE-based PEPS system by adopting six BLE beacons for localization and an additional nine BLE scanners for environment prediction, where the data obtained are collected by a smartphone that acts as a concentrator.

In general, other techniques for indoor localization are based on Ultra-WideBand (UWB) technology; typically, these are used for localization in complex multi-stance environments. This technology involves the utilization of higher frequencies than BLE, which correspond to higher power consumption. As described in [12], the authors achieve a 95% accuracy level of room identification; however, they adopt a total of sixteen anchors and distances to the set of predefined landmarks distributed in the indoor environment as the input parameter for the neural network.

In this paper, we focus attention on improving the accuracy of a driver’s localization with respect to the vehicle; in particular, Bluetooth Low-Energy is used to create a connection between man (through the smartphone) and machine by creating a PEPS system. This connection occurs thanks to the use of six ESP32s, 32-bit micro-controllers with integrated antennae, mounted inside the passengers’ compartment. In the proposed architecture, the ESP32s act as BLE scanners and are mounted according to a network topology, called Ψ network, suitably designed and tested for this research activity to improve the overall performance of the method; in contrast, the driver’s smartphone acts as BLE advertiser [13]. One of the six ESP32s has the task of classifying the smartphone’s location using the k-NN (k- Nearest Neighbour) classifier [14]. The BLE Fingerprinting positioning technique was based on RSSI (Received Signal Strength Indicator) [14] acquisition, successively filtered by means of a proper Kalman Filter to further improve Fingerprinting and classifier performance [15]. Finally, particular attention has been paid to the spatial orientation of each BLE scanner in order to grant the highest value of RSSI.

The article is organized as follows; the implementation of the PEPS system is described in Section 2, while in Section 3 hardware set-up and obtained results are presented before drawing the conclusions in Section 4.

## 2. Proposed Method

This section will describe the solution adopted for the implementation of the PEPS system (Figure 1). Hardware characteristics of the adopted microcontroller will be first described, followed by methodologies and techniques exploited to overcome the problems highlighted in the literature. Since GPS has localization limitations in scenarios involving closed environments or structures, possible choices lead to the use of Wi-Fi or BLE. Previous work suggests that the main limitation for Wi-Fi is router dependency, the inability to install it in many environments and the attenuation related to the distance from it, while for the BLE the established connection depends exclusively on the distance between the reference point and user in all environments.

### 2.1. Positioning Algorithm

Several problems arise in indoor positioning related to the complexity of the context, such as the multipath effect [16], the attenuation and dispersion of the signal due to the density of obstacles, the problem of Non-Line of Sight (NLoS) transmission of the signal of interest. Different ranging methods are used to determine the position of an emitter given a series of reference points of known coordinates. The methods differ in accuracy, hardware and operating logic. To obtain the coordinates of the emitter, the time elapsed between emission and reception and the direction from which the signal comes can be exploited. The technique used to determine distance information is Received Signal Strength Indicator (RSSI), which is based on the received signal strength and the relationship between the attenuation of the signal and the traveled distance. To obtain a good localization, a preventive analysis of the environment is required in order to store some information, with which the data obtained during the detection of the position will be compared or by applying the Fingerprinting technique.

Once distance information has been obtained with ranging methods, it is necessary to process this data in order to transform distances, times and signal strength into coordinates and, consequently, the position. The technique used to determine the position is the *Fingerprinting* technique, usually applied on measured values of RSSI. It consists of detecting the BLE signals present in a given area in order to obtain a “Fingerprint” available to one or more devices for their localization; the position will be calculated using a cluster-matching algorithm, i.e., by finding, among all the previously saved detections, the one that best approximates the detection made at the run-time by the device. The adopted Fingerprinting algorithm [14] consists of two successive phases (Figure 2):Offline phase, where the environment is analyzed using a grid of points with respect to which groups of RSSI values will be associated with the position;Online phase, the groups of RSSI values acquired in the offline phase and those acquired in real-time will provide the most reliable position of the object.

**Figure 2 sensors-22-09615-f002:**
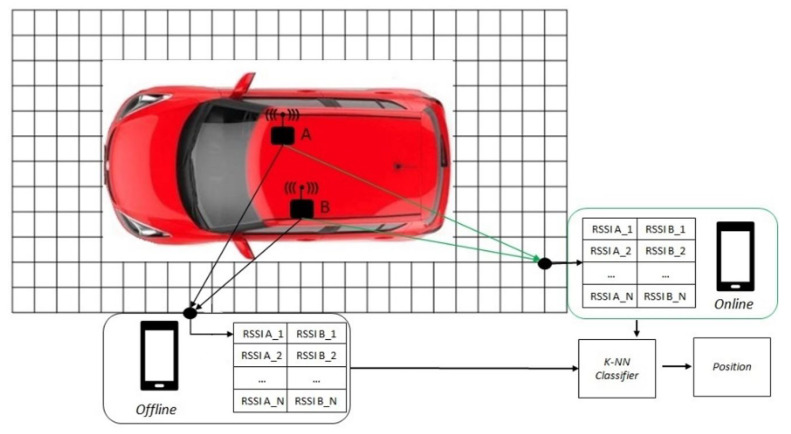
Localization system using BLE Fingerprinting.

Although it requires a longer implementation time and a high computational cost, it has proven to provide excellent results.

### 2.2. Filtering Algorithms

Due to the effects described above, the RSSI values exhibit random and unstable behavior, generating noise, also when both reference and target remain stationary in space. In order to reduce as much as possible the uncertainty sources, different filtering algorithms were taken into account, in both pre-processing and post-processing stages:Moving average filter (MA) [17], based on the collection of n RSSI samples and calculating the average of the values to obtain a smoothed value;Median filter, based on the calculation of the median of n RSSI samples, dividing the ordered sequence of values into two sequences of an equal number of terms, one having a lower value and the other having a higher value than the median itself; the filter proves to be an optimal solution to remove outliers in the measured data;Kalman Filter (KF), operating according to a prediction and correction approach. In the prediction stage, the Kalman Filter provides an a priori estimate of the state vector starting from that obtained in the previous step. The estimate is then improved in the update stage thanks to the availability of some measures, thus achieving the so-called a posteriori estimate, exploited as input in the successive cycle.

The best performance was achieved by means of the Kalman Filter; from an operative point of view, the Kalman Filter’s operating steps are described by means of the following expressions:

Prediction:Prediction of the current state:
(1)$$x_{k}=A\times x_{k-1}$$Prediction of the covariance of the error:
(2)$$P_{k}=A\times P_{k-1}\times A^{T}+Q$$

Update:Calculation of the gain of Kalman:
(3)$$K=P_{k}\times H^{T}\times {\left(H\times P_{k}\times H^{T}+R\right)}^{-1}$$Calculation of the new state:
(4)$$x_{k}=x_{k-1}+K\times \left(Z_{k}-H^{T}\times x_{k-1}\right)$$Calculation of the new covariance of the error:
(5)$$P_{k}=P_{k-1}-\left(K_{k}\times H\times P_{k}\right)$$ where:$x_{k}$: Estimate of the current state.*A*: State transition matrix.$P_{k}$: Average error estimate for the current state.*Q*: Estimated covariance of process errors.*K*: Kalman gain.*H*: Observation matrix.*R*: Estimated covariance of the measurement error.$Z_{k}$: Measurement vector for the current state.

As for the considered application, the identity matrix was chosen for both matrices H and A while $Z_{k}$ corresponds to the *k-th* sample of the measured RSSI signal. The considered settings were chosen due to the reduced dynamic of the considered parameter.

### 2.3. Classification: K-NN

The *k-Nearest Neighbor* algorithm classifies an unknown sample considering the class of the k closest samples of the training set. To create a k-NN type classifier, it is necessary to calculate the distances between the sample to be classified and all the training samples, identifying the k-closest training samples and the respective label. Finally, the algorithm compares the label of these k points that are the closest to our sample. The destination label with the highest frequency among these k-points is assigned as the destination class to the new sample. This classifier lends itself well to the BLE Fingerprinting technique, as the samples obtained from the online phase are compared by likelihood with the training samples, represented by the radio map of the points obtained from the offline phase. The main advantages of this method are that it does not require learning or building a model, it can adapt its decision boundaries arbitrarily, producing a more flexible model representation and also guaranteeing the possibility of increasing the training set.

### 2.4. Network Topology

The study of network topologies allows the definition of the position of all the nodes that are part of the network and their connections, both physical and logical. Using graph theory, the most important parameters such as the number of nodes, the number of transmission channels and redundancy are taken into account, keeping fault tolerance under control [18]. In order to correctly locate the position of the smartphone with respect to different vehicle zones, different network topologies were tested. In particular, attention has been focused on star and tree networks; the results obtained by these analyses were absolutely unreliable and with them it was not possible able to distinguish most of the different vehicle zones. To achieve better performance, it was decided to adopt a “hybrid” network called the Ψ network (Figure 3), which has the same layout as the tree network but functions in the same way as the star network where the top central node represents the star center, responsible for receiving, acquiring and processing data. The adoption of this hybrid network allows the recognition of the smartphone’s position related to the different vehicle zone with a high degree of reliability.

## 3. Result and Discussion

As stated above, the Kalman Filter was used as the filter algorithm to improve RSSI estimates, while Fingerprinting was exploited as the technique to determine the driver’s position. The implementation and assessment of the proposed method passed through five main steps:Orientation in the space of the ESP32, in order to establish the one that best discriminates the position in which the smartphone is located [19].Implementation of the complete system, consisting of six ESP32s which are interested in the external and internal position of the smartphone, in order to create and store the radio map of the points.Real-time acquisition and classification of the smartphone position.Processing of the data acquired in phase III through statistical analysis, such as ANOVA (Analysis of Variance) and Siegel–Tukey Test.Evaluation of the classifier’s accuracy using the Classification Learner app in the Matlab environment, carrying out training, validation and testing on the acquired data.

### 3.1. ESP32 Microcontroller

ESP32 microcontroller by Espressif Systems [20] has a built-in antenna supporting both Wi-Fi and Bluetooth 2.4 GHz equipped with 48 GPIO pins, belonging to the low-cost device range. It implements different types of protocols such as TCP/IP, MAC WLAN, Wi-Fi and Bluetooth; moreover, it provides UART, SPI, I2C, I2S and Capacitive Touch interface. Its high performance, robustness and versatility make it ideal for a wide range of applications, such as wearables and IoT. The dynamic scaling of the power, due to the low duty cycle, allows for minimization of the dissipated energy as long as there is a suitable trade-off between communication range and data transfer speed. State-of-the-art performance through digital calibration provides +20.5 dBm of average power. The proposed hardware system, consisting of six ESP32s, overcomes the intrinsic limit of beacons, which can only be used in Advertiser mode and cannot perform other actions, such as low energy consumption and data processing.

### 3.2. ESP32 Orientations

The first step aims at determining the optimal orientation of the ESP32. To this aim, two measurement campaigns were carried out to evaluate which configuration, either ESP32 as a scanner and smartphone as an advertiser or vice versa, should be preferable for the purpose.

Moreover, by rotating the ESP32 with respect to its z- and y-axis (Figure 4), both campaigns were carried out in sub-configurations:ESP32 located inside the vehicle at the driver’s side and smartphone positioned either externally in four positions, i.e., driver side or left (LoS), passenger side or right (NLoS), front side (NLoS) and backside (NLoS) or internally in two positions (both NLoS), i.e., driver’s seat and passenger’s seat (Figure 5a).ESP32 located inside the vehicle at the driver’s side and smartphone positioned internally in two positions (both NLoS), i.e., driver’s seat and passenger’s seat (Figure 5b).

**Figure 4 sensors-22-09615-f004:**
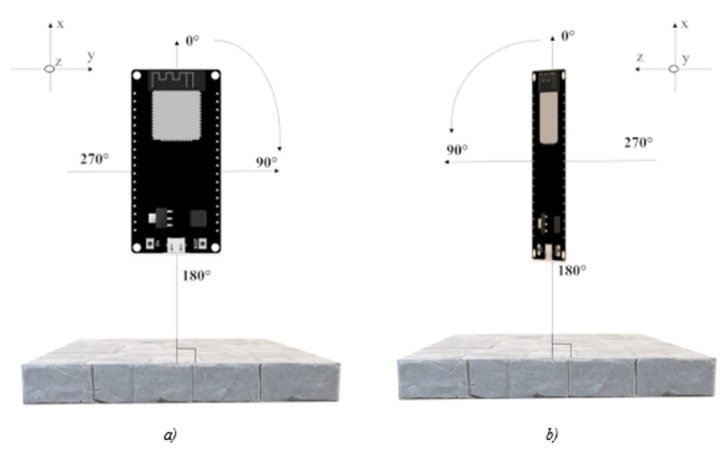
ESP32 orientation: (**a**) Rotation with respect to the z-axis. (**b**) Rotation with respect to the y-axis.

**Figure 5 sensors-22-09615-f005:**
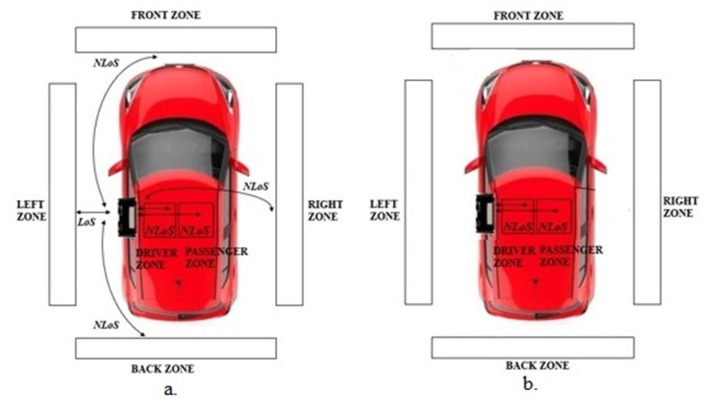
LoS and NLoS for different zones: (**a**) Configuration 1. (**b**) Configuration 2.

In both cases, measured values of RSSI were processed and filtered by means of a Kalman Filter that ensures the reduction of the RSSI measurement noise sources. In order to choose the best orientation of ESP32, a loss function was evaluated that could relate the average and standard deviation of the distributions of RSSI values and that could search for the maximum distance between the centers of the distributions that have a small standard deviation:(6)$$f_{loss}=\frac{\mu_{SX}-\mu_{DX}-\mu_{ANT}-\mu_{POS}}{\sqrt{\sigma_{SX}^{2}+\sigma_{DX}^{2}+\sigma_{ANT}^{2}+\sigma_{POS}^{2}}}$$
(7)$$f_{loss}=\frac{\mu_{DRV}-\mu_{PASS}}{\sqrt{\sigma_{DRV}^{2}+\sigma_{PASS}^{2}}}$$
where:$\mu_{SX}$, $\mu_{DX}$, $\mu_{ANT}$, $\mu_{POS}$, $\mu_{DRV}$, $\mu_{PASS}$ (dB): average of the different zones calculated from the respective arrays of RSSI values;$\sigma_{SX}$, $\sigma_{DX}$, $\sigma_{ANT}$, $\sigma_{POS}$, $\sigma_{DRV}$, $\sigma_{PASS}$ (dB): standard deviations of the different zones calculated from the respective arrays of RSSI values.

The best results were associated with configuration 1 and, in particular, with the orientation 180${}^{\circ }$–90${}^{\circ }$, which presents a significant value of loss function as well as better management, in terms of software and realization of the complete system. As can be appreciated from Table 1, the loss function values are somewhere greater than those associated with the chosen orientation; unfortunately, those orientations were characterized by unsuitable features as a non-Gaussian data distribution, overlapping of the averages of the different zones and a high standard deviation that does not comply with the selection criteria.

### 3.3. System and Radio Map Realization

Once the orientation of the ESP32 was defined, the localization performance of the PEPS-BLE system was assessed, using a Suzuki Swift as a test vehicle and a Huawei P10 Lite smartphone representing the smart key. The size of the test environment, including the vehicle, is 4.0 m by 6.0 m. For the offline phase of the BLE-Fingerprinting model, a radio map was created consisting of 90 reference points, each of which is spaced 50 cm for the external areas and 15 cm for the internal areas. The coordinates of the test environment were divided into six different areas (1—left area; 2—front area; 3—right area; 4—rear area; 5—driver’s area; 6—passenger’s area), each consisting of fifteen points represented by the intersection of the sides of the rectangles, as in Figure 6.

Once the radio map was defined, the complete system was assembled following the layout defined by the chosen network topology (Figure 7). During the offline phase, RSSI values measured by each ESP32 were transmitted according to the Universal Asynchronous Receiver-Transmitter (UART) protocol (Figure 8); in particular,
ESP32-E, ESP32-F and ESP32-D acted as Transmitter, sending their acquired data to ESP32-A, ESP32-C and ESP32-B, respectively.ESP32-A and ESP32-C acted as Receiver-Transmitter, forwarding the data received from the leaves nodes and sending their own RSSI measures to the ESP32-B.ESP32-B acted as a concentrator, receiving and processing its measured RSSI along with the other received values.

**Figure 7 sensors-22-09615-f007:**
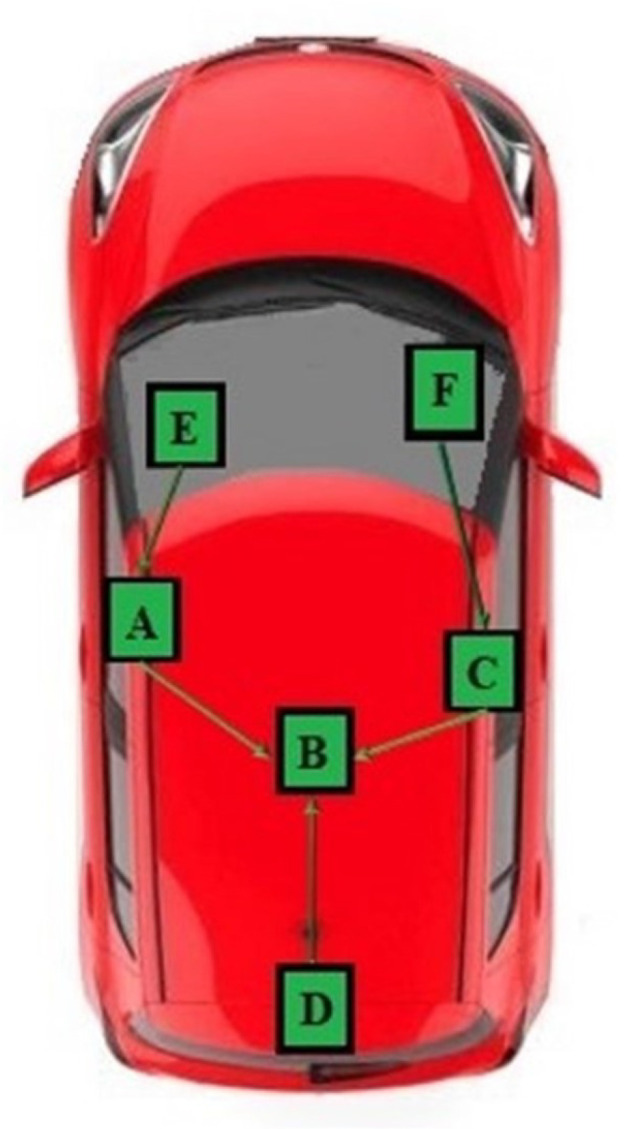
Layout of network topology with ESP32.

**Figure 8 sensors-22-09615-f008:**
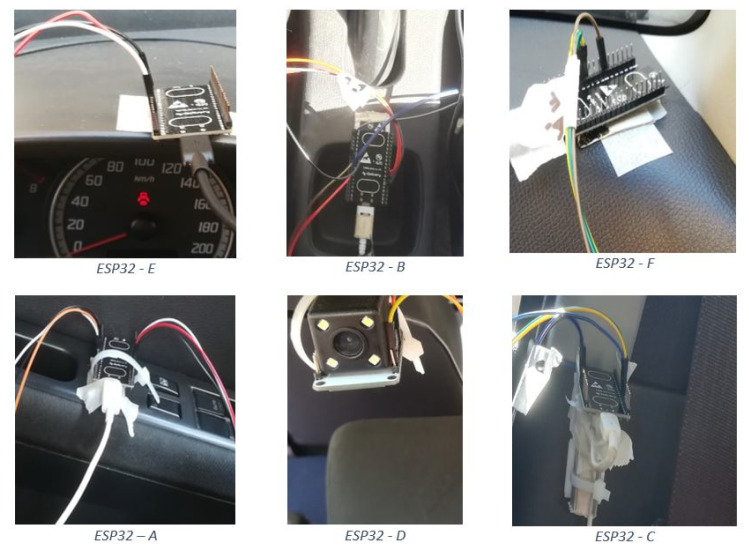
Definition of roles for every ESP32.

### 3.4. Offline and Online Data Acquisition

During the offline phase, several RSSI values were acquired and filtered for each different position of the grids in Figure 6. In particular, 130 samples were acquired for each point of the grid with a sampling rate of 1 Hz; acquired samples were averaged and finally arranged according to a 15X6 matrix containing the mean values of RSSI of each grid position of each zone. These average values were exploited for the training phase of the k-NN classifier. In the online phase, for each zone, the acquisition took place in a time interval equal to 360 s–420 s.

### 3.5. Statistical Inferences

The significance of the differences experienced in RSSI measures has been assessed by means of statistical analysis such as One-Way ANOVA and the Siegel–Tukey Test. In both analyses, the zones are associated as described in the following:The Left zone;The Front zone;The Right zone;The Rear zone;The Driver zone;The Passenger zone.

#### 3.5.1. One-Way ANOVA

This section shows the results of the ANOVA test for the radio map (Figure 9) and for the data obtained in the online phase as a result of the classifier (Figure 10). In particular, the results are shown in terms of box plot; the rectangle is delimited by the first and third quartiles, divided by the median, while the external segments corresponded to the minimum and maximum encountered values. In this way, the four intervals that were equally populated and that corresponded to the quartiles are graphically represented.

As can be appreciated from Figure 9 and Figure 10, the different positions of the driver (either inside or outside the vehicle) are significantly distinguished.

#### 3.5.2. Siegel–Tukey Test

Once ANOVA determined the significance of the different values of RSSI, the Siegel–Tukey Test was performed to evaluate the correlation of the RSSI value for every zone where the ESP32s are placed. To evaluate the classifier performance, the Siegel–Tukey Test was adopted. The result of the Siegel–Tukey Test for the radio map before the processing is shown in Figure 11, which the highlighted zones Left, Right and Rear overlap, i.e., the different positions could not be identified. Instead, the Siegel–Tukey results, after the filtering and classification process (Figure 12), highlight that the different zones are clearly marked and it is possible to identify the correct zone.

The Tukey test for the radio map shows both visually and statistically that the group averages for the internal and external zones are significantly different in terms of significance α (equal to 0.05), expressing the potential of the system to recognize if the smartphone is inside the vehicle or outside. It is also evident, however, that the averages for the external zones and for the internal zones are not significantly different.

### 3.6. Classification Learner

The Classification Learner was useful in training the models exploited to classify the data. The dataset consists of columns representing the values acquired by the six ESP32s, the relative classification by the ESP32-B which has the task of classifying and the expected classification relating to the area to which they belong. The dataset (2208 × 8) was imported and reorganized, randomly arranging the matrix rows. The data were divided into 70% for the training phase and 30% for the testing phase. As for the results, it was possible to observe:The Scatter Plot (Figure 13), which helps to examine the characteristics to be included or excluded, to visualize the training data and the points classified incorrectly, sensing a dispersion of data such as to believe that the system has encountered some difficulties only in certain areas where it creates a certain overlap of samples; in fact, as shown in Figure 13, the classifier is able to identify the RSSI values for every zone with an accuracy of 98.7 %, while the system presents incorrect predictions only for a few values between Zones 1 and 4.The Confusion Matrix (Figure 14) identifies the areas where the classifier was accurate. In fact, the Positive Predictive Value (PPV) and False Discovery Rate (FDR) are defined as the proportion of correctly classified observations per expected class and the proportion of incorrectly classified observations per intended class, respectively (Figure 15). In contrast, the True Positive Rate (TPR) and False Negative Rate (FNR) defined as the proportion of correctly classified observations per real class and is the proportion of observations incorrectly classified per true class, respectively, (Figure 16) [21]. Comparing the true class with the predicted class, noting the number of observations correctly evaluated, the predictive positive rates are greater than the false negatives and false detection rates, so the classifier method adopted is capable of identifying with a high degree of reliability the correct user position in the different zones tested.

In order to quantify the agreement between measured and true values of the measure, the accuracy was evaluated according to [22]:(8)$$ACCURACY=\frac{TP+TN}{TP+TN+FP+FN}$$
where *TP* is the true positive, *TN* is the true negative, *FP* is the false positive and *FN* is the false negative.

In particular, the overall accuracy reached by the proposed method thanks to the filtering stage and the exploited network topology was equal to 98.5%, greater than that provided by similar solutions presented in the literature [11].

## 4. Conclusions

The aim of the paper was the definition, implementation and assessment of a model for a PEPS-BLE system based on micro-controllers, as a means of interaction between smartphone and vehicle. Once their optimal orientation is defined in space, the complete system consisting of six micro-controllers in an asymmetrical arrangement, communicating via UART protocol according to the experimental network topology, called Ψ network, was created and installed. By adopting the BLE Fingerprinting technique, consisting of an offline phase for the construction of the radio map and an online phase, for the acquisition of real-time samples and through the use of the K-NN classifier that was previously evaluated by means of the Kalman Filter, the system has granted an overall accuracy of 98.5%, thus allowing to reliably distinguishing whether the user/smartphone is inside or outside the vehicle. As a comparison with other results reached in the literature, in [11] the accuracy obtained was equal to 94,6%, thus confirming the benefit brought by the adoption of a dedicated embedded system to collect data, the selection and implementation of a different topology network (composed by only six BLE nodes) and the Kalman Filter approach proposed that ensures an overall system performance enhancement. The method’s performance was assessed by considering only one smartphone model; future works will include a measurement campaign involving different smartphones in order to carry out a more comprehensive performance assessment and provide the possibility for its deployment as a commercial solution. To this aim, thanks to a suitable connection of the BLE devices with the car ECU, it would be possible to allow actions managed directly from the smartphone, such as unlocking the doors or starting the car, making the user–car relationship smart.

## Figures and Tables

**Figure 1 sensors-22-09615-f001:**
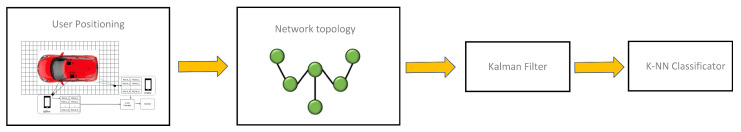
Proposed architecture for the implementation of a PEPS system.

**Figure 3 sensors-22-09615-f003:**
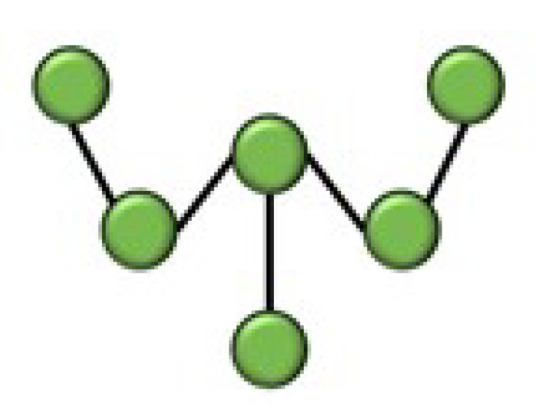
Ψ network topology.

**Figure 6 sensors-22-09615-f006:**
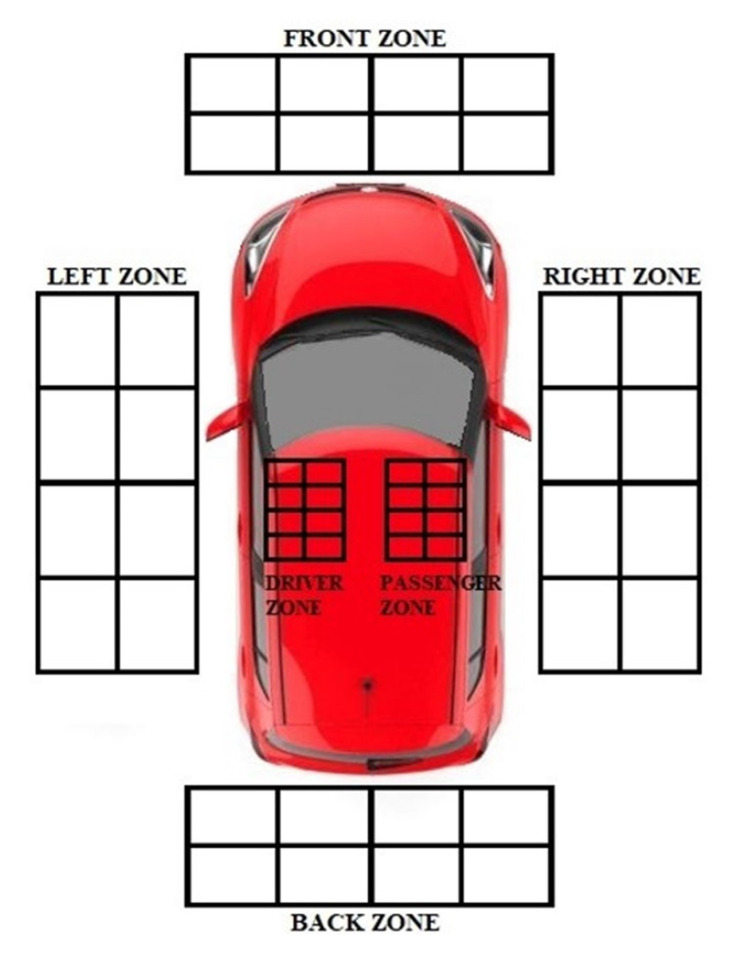
Radio Map.

**Figure 9 sensors-22-09615-f009:**
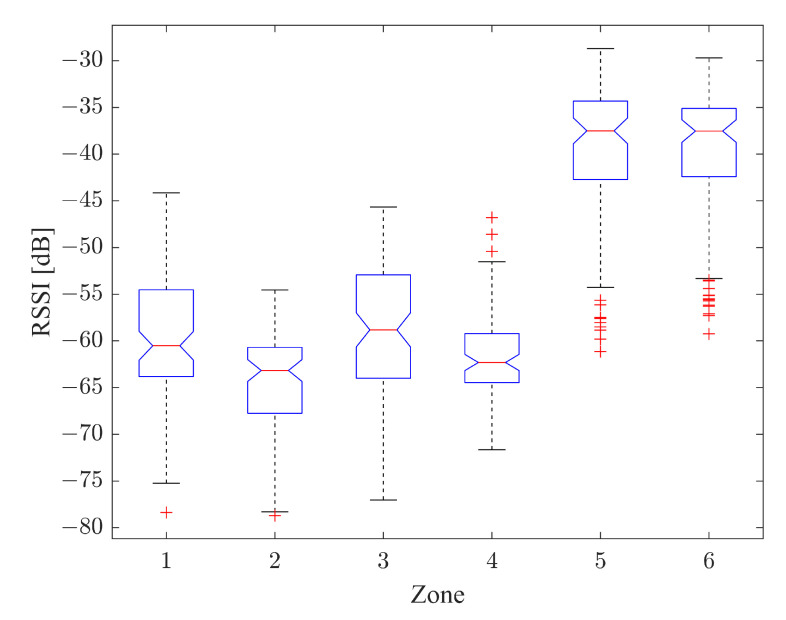
ANOVA Box Plot for Radio Map.

**Figure 10 sensors-22-09615-f010:**
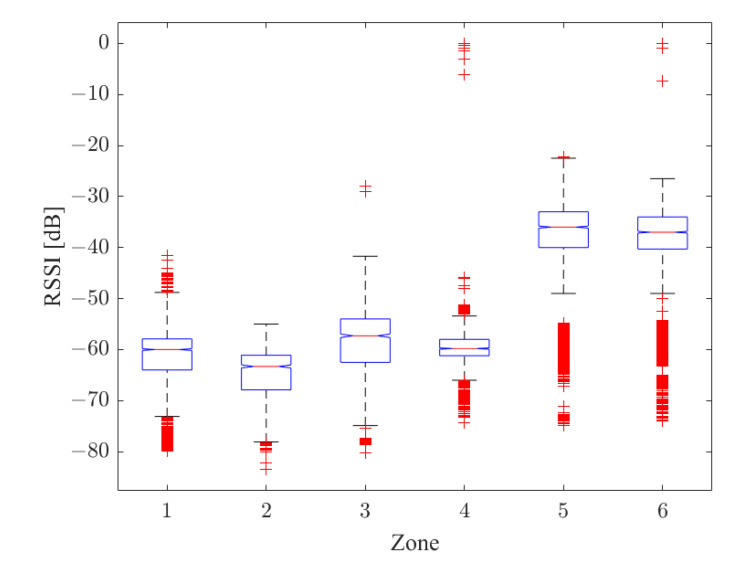
ANOVA Box Plot for classifier output.

**Figure 11 sensors-22-09615-f011:**
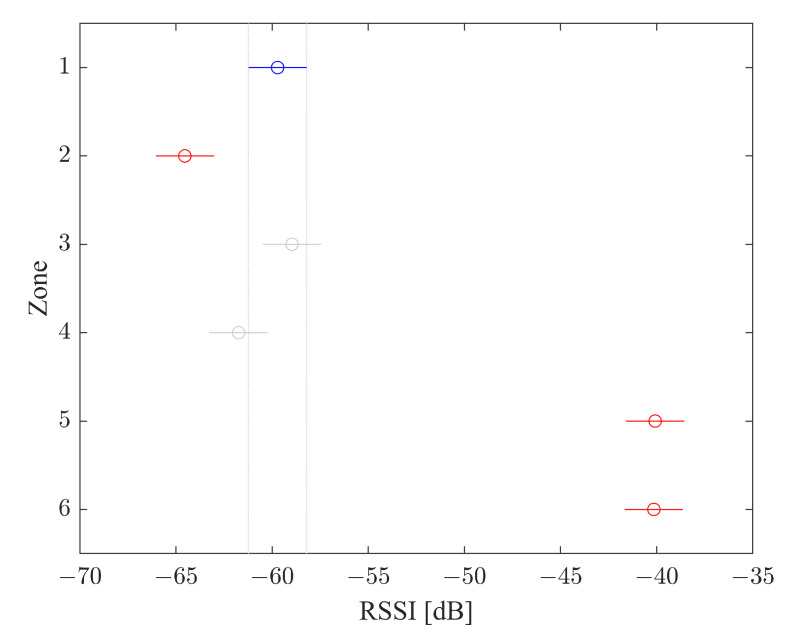
Siegel–Tukey Test for Radio Map.

**Figure 12 sensors-22-09615-f012:**
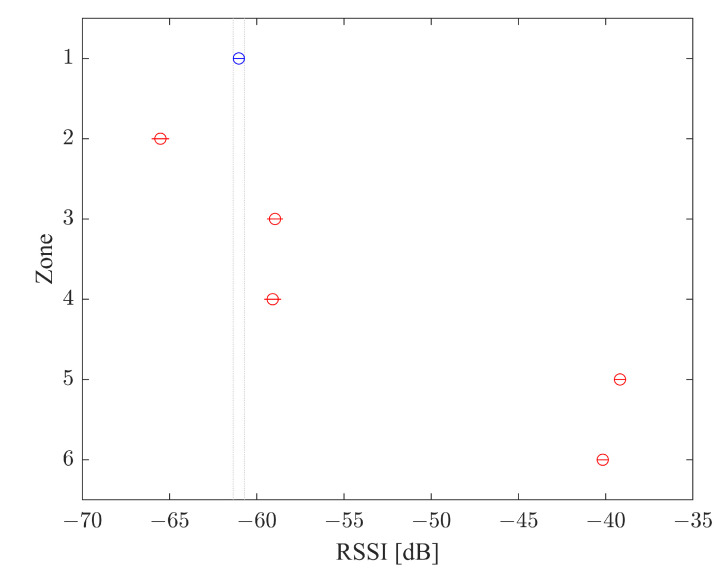
Siegel–Tukey Test for classifier output.

**Figure 13 sensors-22-09615-f013:**
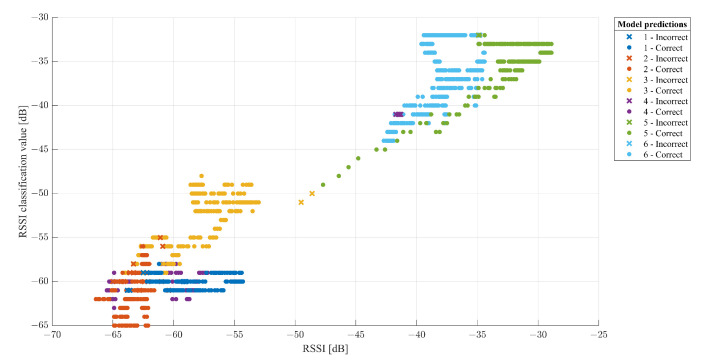
Dataset scatter plot.

**Figure 14 sensors-22-09615-f014:**
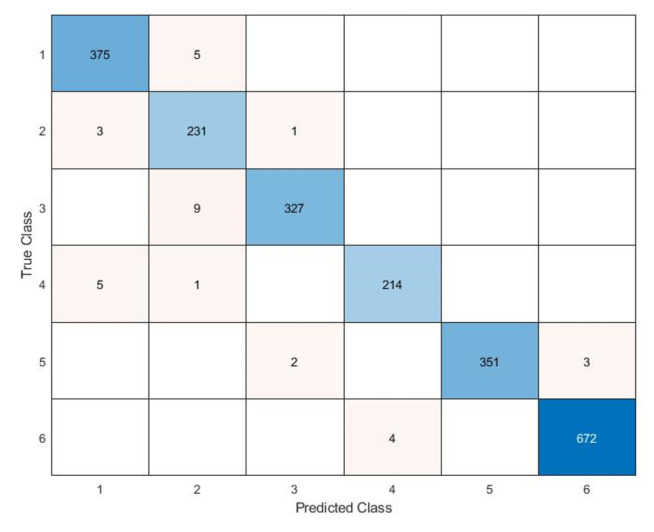
Confusion Matrix: Number of observations.

**Figure 15 sensors-22-09615-f015:**
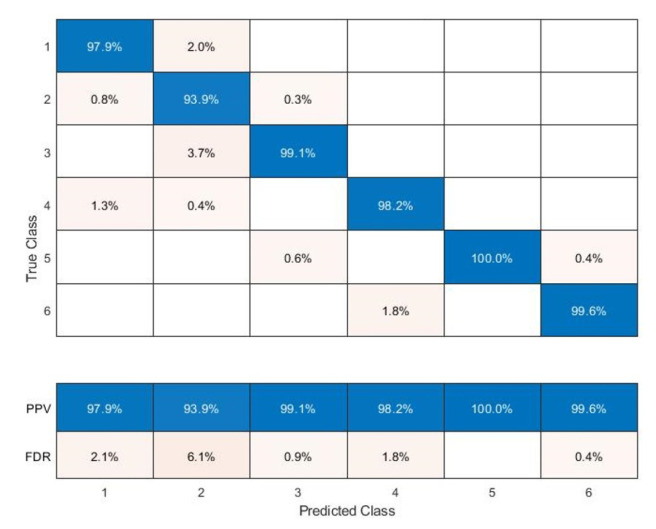
Confusion Matrix: Positive Predictive Values vs. False Discovery Rate.

**Figure 16 sensors-22-09615-f016:**
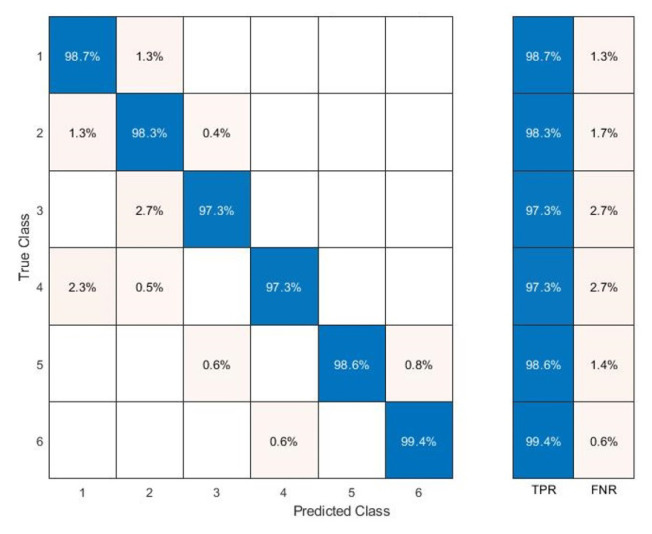
Confusion Matrix: True Positive Rate vs. False Negative Rate.

**Table 1 sensors-22-09615-t001:** Loss function results for ESP32 orientation referring to the two different configurations evaluated as scanner and smartphone as advertiser.

	Configuration 1		Configuration 2	
**Orientation**	**Raw Values**	**Smoothed Values**	**Raw Values**	**Smoothed Values**
0${}^{\circ }$–0${}^{\circ }$	13.0633	26.1039	1.0604	2.9444
0${}^{\circ }$–90${}^{\circ }$	12.5879	23.8130	0.0562	−0.1474
0${}^{\circ }$–180${}^{\circ }$	12.4053	22.6049	1.1373	1.2806
0${}^{\circ }$–270${}^{\circ }$	11.7701	17.4743	−0.2721	−2.1425
90${}^{\circ }$–0${}^{\circ }$	8.9414	16.4160	0.2833	0.2847
90${}^{\circ }$–90${}^{\circ }$	10.4545	17.5767	0.6223	1.5775
90${}^{\circ }$–180${}^{\circ }$	13.5501	23.3651	−0.8943	−1.0363
90${}^{\circ }$–270${}^{\circ }$	10.7310	19.9310	0.5055	3.4045
180${}^{\circ }$–0${}^{\circ }$	11.7475	32.4822	0.8637	1.3109
180${}^{\circ }$–90${}^{\circ }$	12.2626	27.5023	0.1383	0.2263
180${}^{\circ }$–180${}^{\circ }$	14.7112	27.5250	1.2878	4.2162
180${}^{\circ }$–270${}^{\circ }$	13.8268	25.4591	0.7454	2.3117
270${}^{\circ }$–0${}^{\circ }$	9.6949	18.2563	−0.0980	−0.3246
270${}^{\circ }$–90${}^{\circ }$	12.6398	22.4502	0.5281	0.7525
270${}^{\circ }$–180${}^{\circ }$	12.5429	20.6201	−0.2232	−1.7390
270${}^{\circ }$–270${}^{\circ }$	11.9059	19.7627	0.2115	1.8816

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
