# Peer review of "An Improved Method Based on Bluetooth Low-Energy Fingerprinting for the Implementation of PEPS System"

_sensors, 2022, doi:10.3390/s22249615_

Round 1

Reviewer 1 Report

For a journal paper readers expect at least a few different smartphones (beacons) since the fingerprinting aspect requires RSSI values and the device RSSI values, as well as the positioning of the BT antennas may vary. I belive 3 devices should be minimum at the very least for readers to consider the results generalizable.

In addition, different topographies should be tested to see if they have any effect on the overall performance in any way.

Author Response

We are very grateful to you for giving us an opportunity to revise and improve our manuscript. We appreciated your positive and constructive comments and suggestions on the manuscript entitled "An Improved Method Based On Bluetooth Low Energy Fingerprinting for the Implementation of PEPS System" (ID: sensors-2035695).

We have studied the reviewers' comments carefully and tried our best to revise our manuscript according to the comments. Responses and revisions are given in the attached file.

Reviewer 2 Report

The authors proposed an improved method for PEPS system implementation. While the paper is interesting, this reviewer has the following concerns that should be addressed before the possible publication.

1.      In the Abstract and Conclusion section, please describe the advantages of your proposed work quantitively as well. For example, you mentioned the global accuracy (which is very good), so I suggest mentioning few more parameters. (Then it would be easier for the reviewers and the readers to distinguish your novel work from the others.)

2.      Please briefly describe the drawbacks of your proposed idea. Please also briefly state your future works (for tackling those shortcomings).

Author Response

(The authors gave the same response as above.)

Reviewer 3 Report

The paper deals with an interesting topic of implementing Bluetooth technology to allow Passive Entry Passive Start in vehicles. 

The title of the paper suggests the proposed method is improved, therefore, the authors should provide some comparison with the original method. 

Comparison with other solutions should be provided. Would it be possible to use fewer nodes implemented in the car and still achieve good classification accuracy? 

Some figures in the manuscript are very low quality, especially confusion matrixes. The authors should improve the quality of the figures. 

How does the proposed solution compare to solutions based on other technologies, for example, based on UWB?

Data from figure 13 are not quite clear, what does "incorrect" mean? 

Does "4 - incorrect" mean it should have been classified as 4 or that the sample was classified as 4 but should have been classified as something else?

Were all the measurements taken under the same conditions, i.e. temperature, humidity, environment, etc.? 

How would a change in environmental parameters affect the accuracy of the system?

Author Response

(The authors gave the same response as above.)

Round 2

Reviewer 1 Report

I'd like to thank the authors for their response.